# The impact of dual- versus single-dosing and fatty food co-administration on albendazole efficacy against hookworm among children in Mayuge district, Uganda: Results from a 2x2 factorial randomised controlled trial

Eun Seok Kim[1,2]*, Moses Adriko[3], Wamboko Aidah[3], Kabarangira Christine Oseku[4], David Lokure[5], Kalpana Sabapathy[1‡], Emily L. Webb[1‡]

**1** Department of Infectious Disease Epidemiology, London School of Hygiene and Tropical Medicine, London, United Kingdom, **2** World Vision Korea, Seoul, Korea, **3** Vector-borne and NTDs Control Division, Ministry of Health, Kampala, Uganda, **4** Save the Children International, Moroto, Uganda, **5** Information and technology sector, Kotido district local government, Kotido, Uganda

‡ These authors are joint senior authors on this work.
* sstone99@gmail.com

## Abstract

### Background

Mass Drug Administration (MDA) is the main strategy for control of soil-transmitted helminth (STH) infections, with single-dose benzimidazole (albendazole or mebendazole) the principal MDA option. In Mayuge district, Uganda, an MDA programme has been in place for over fifteen years but hookworm infection remains common and there is concern that the effectiveness of single-dose albendazole as currently used for MDA may be sub-optimal. This study aims to assess the efficacy of dual- versus single-dose albendazole, with and without fatty food co-administration against hookworm, the dominant form of STHs in Mayuge district, Uganda.

### Methodology

This was a 2x2 factorial randomised controlled trial to investigate two interventions simultaneously; 1) dual-dose versus single-dose albendazole, 2) taking albendazole with or without fatty food (200 grams of avocado eaten directly after medication). School children with hookworm infection were randomised in a 1:1:1:1 ratio to the four possible treatment groups. Three weeks after the treatment, stool samples were collected from trial participants to evaluate trial outcomes: cure rate and egg reduction rate (ERR).

### Principal findings

A total of 225 participants were enrolled, and 222 (98.7%) seen at 3 weeks. The cure rate in the dual-dose group was 96.4% (95% CI: 90.9–99%), higher than 83.9% (95% CI: 75.7–90.2%) in the single-dose group (OR: 5.07, 95% CI:1.61–15.96, $p = 0.002$). The ERR was

**Data Availability Statement:** All relevant data are within the manuscript and its Supporting Information files (S1 Data and S1 Code).

**Funding:** Korea International Cooperation Agency (KOICA) financially supported the MANE project, and this article was published in collaboration with KOICA and World Vision (Grant No: 2019-04). World Vision Korea received the funds from KOICA via the Global Diseases Eradication Fund. Dr. Eun Seok Kim was the health specialist and principal investigator of the MANE project. This study was conducted as part of the MANE project. The funders had no role in the study design, data collection and analysis, decision to publish, or preparation of the manuscript.

**Competing interests:** The authors have declared that no competing interests exist.

97.6% and 94.5% in the dual-dose group and single-dose drug group, respectively (ERR difference 3.1%, 95% CI: -3.89–16.39%, $p$ = 0.553). The cure rates among participants taking albendazole with and without avocado were 90.1% and 89.1%, respectively, with no statistical difference between the two groups (OR: 1.24, 95% CI: 0.51–3.03, $p$ = 0.622). The ERR was 97.0% and 94.2% in the group receiving albendazole with and without avocado, respectively, and the difference in ERR between the two groups was 2.8% (95% CI -8.63–14.3%, $p$ = 0.629).

## Conclusions/significance

In Ugandan school children, dual-dose albendazole improves the cure rate of hookworm compared to single-dose albendazole. However, there was no significant improvement in cure rate or egg reduction rate of hookworm with fatty-food co-administration. Dual-dose albendazole is a feasible alternative for improving drug effectiveness against hookworm infection and minimising drug resistance.

## Trial registration

PACTR202202738940158.

### Author summary

Mass Drug Administration (MDA) of single-dose albendazole is the main strategy for control of soil-transmitted helminth infections including hookworm, but these infections remain common in many settings where MDA has been implemented for several years. This study aimed to assess the efficacy of dual-dose versus single-dose albendazole, with and without fatty food co-administration, against hookworm infection in Mayuge district, Uganda. The study was conducted as a randomized controlled trial among school children with hookworm infection. The cure rate was found to be significantly higher in the dual-dose group compared to the single-dose group, with no significant difference in the cure rate or egg reduction rate observed between the groups that took albendazole with or without fatty food. The study suggests that dual-dose albendazole is a feasible alternative for improving drug effectiveness against hookworm infection and minimizing drug resistance. Although the study did not demonstrate any benefit with the use of avocado co-administration against hookworm infection, further exploration is warranted due to the biological rationale for improving albendazole efficacy to treat intraluminal hookworm infection.

## Introduction

Soil-transmitted helminth (STH) infection, also known as intestinal worm infection, is classified as a neglected tropical disease (NTD) because while it is very common, it remains neglected globally [1,2]. There are three dominant STHs: roundworm (*Ascaris lumbricoides*), whipworm (*Trichuris trichiura*) and hookworm (*Ancylostoma duodenale* and *Necator americanus*) [3,4], with *Strongyloides stercoralis* another STH not frequently identified in the standard stool test. Hookworm is the most prevalent STH causing anaemia, contributing to approximately 4 million DALYs (95% uncertainty interval: 395,922–16,499,971) globally per

year [5]. STH primarily affects the world's deprived populations, causing significant health and socio-economic repercussions and constituting an important public health problem in developing countries. The World Health Organization (WHO) estimates that STH affects more than 1.5 billion people worldwide [6]. Globally, the prevalence of hookworm and *Ascaris lumbricoides* is ~13.6% for both species, while the prevalence of *Trichuris trichiura* is ~11.6% in sub-Saharan Africa [6].

Preventive chemotherapy has been the backbone of STH control strategy. However, effective control requires improved access to safe water, adequate sanitation, vector control, and health education to accelerate progress towards elimination [7–9]. The WHO recommends annual or biannual treatment with benzimidazole drugs (400mg of albendazole for adults and children over two years, 200mg of albendazole for children aged one to two years or 500mg of mebendazole) against STHs, for pre-school-aged children (one to four years old) and school-aged children (five to 14 years old) living in areas where the prevalence of STH is over 20% [10].

In Uganda, a nationwide school-based prevalence survey was conducted in 2004 including 20,185 children (aged 5–20 years) from 207 schools, and the prevalence of STHs was 54.8% [11], which reduced to 8.8% in a 2016 nationwide school-based survey which included 4,275 children from 120 schools [12]. Hookworm was the dominant type of STH in both the 2004 and 2016 studies with 43.5% and 7.7% prevalence, respectively. However, despite the overall reduction between 2004 and 2016, the prevalence of hookworm increased in certain areas. For example, the prevalence in Mpigi district increased from 4.9% to 11.6%, and in Kaliro district increased from 16.4% to 21.9%. This is despite MDA being implemented in Uganda since 2003. Among the 2016 nationwide survey participants with hookworm infection, most (97.1%) had light intensity infections (defined as 1–1,999 Eggs Per Gram) [12].

Proposed theories for the continued high prevalence of hookworm infection in the context of MDA include the possibility that annual single-dose albendazole used for MDA is not sufficient to effectively treat hookworm due to lack of absorption of albendazole [13,14]; variation in drug effectiveness from different brands of albendazole [15,16]; albendazole resistance; lack of MDA coverage for specific marginalized groups such as low socioeconomic status, minority religion, and minority tribes [17]; or that the effect of annual monotherapy with benzimidazole drugs has been handicapped by intrinsic environmental and social factors [18–21]. Lack of access to safe water and sanitation continues to be an overarching factor in hampering efforts to reduce hookworm infection through MDA, due to the continued risk of re-infection [22,23]. While the reported drug effectiveness of albendazole for hookworm treatment is highly variable [24], double or triple doses of albendazole administration show higher drug effectiveness than single-dose therapy [25–28]. Additionally, it has been shown that the maximum active metabolite of albendazole, albendazole sulphoxide concentration (Cmax), was 6.5 times higher in human plasma when albendazole was provided with a fatty food compared to single-dose albendazole alone [14]. Current practice for treating STH in Uganda is to use single-dose albendazole in MDA programmes. The purpose of this trial was to assess the efficacy of dual- versus single-dose albendazole, and the effectiveness of fatty food co-administration using avocado versus no fatty food co-administration, against hookworm, the dominant form of STHs in Mayuge district, Uganda.

## Methods

### Ethics statement

The Ugandan Vector Control Division Research Ethics committee (VCDREC; reference number: UG-REC-018/VCDREC104), Uganda National Council for Science and Technology

(UNCST; reference number: HS1411ES), National Drug Authority (NDA; reference number: CTC 0202/2022) and the London School of Hygiene and Tropical Medicine ethics committee (LSHTM ethic reference:25818) approved the trial. The trial is registered in the Pan African Clinical Trial Registry (registry number: PACTR202202738940158). The authors designed, recorded and reported this trial according to GCP standards. In addition, a designated monitoring expert approved by Ugandan NDA monitored this trial to ensure that GCP standards were met. Written assent forms were obtained from all participants, along with written consent forms from their guardians.

This trial was embedded within the framework of the Mayuge NTD Elimination (MANE) project funded by Korean International Cooperation Agency (KOICA). A prevalence survey was undertaken in January-February 2022 to evaluate the impact of the MANE project implemented between 2019 and 2021. Children who were identified as hookworm positive in the 2022 prevalence survey were invited to take part in the trial. We followed the CONSORT checklist to report this trial (S1 Checklist). The clinical trial was conducted following the procedures outlined in the protocol document (S1 Protocol).

## Study area

Mayuge district is located in south-eastern Uganda, next to Lake Victoria and 146 km from Uganda's capital, Kampala. It has 13 sub-counties. According to the results of the MANE project registration activity in 2020, Mayuge's population is 509,118; with 95,418 children under five; and 156,182 children between five and 14 years old. It has a total of 504 primary schools, 41 health centres, and one hospital (Buluuba hospital).

## Study participants, inclusion and exclusion criteria

In Mayuge district, primary four (P4) and primary five (P5) students aged 8–16 years were the target population for the prevalence survey in January and February 2022. Target schools were selected with probability proportional to school size and an equal number of students from each selected school were invited to participate. English and Lusoga (the local language in Mayuge district) versions of the consent form and child assent form were used.

Among children who were detected as having hookworm infection in the prevalence survey in January and February 2022, randomly selected hookworm positive students were invited to participate in a randomised controlled trial to assess the impact of dual- vs single-dose albendazole and the impact of co-administration with avocado (a fatty food) versus no co-administration with avocado, on hookworm cure rate and egg reduction rate.

Trial inclusion criteria were

1. Child identified with hookworm infection in the January and February 2022 prevalence survey

2. Child attending randomly selected P4 and P5 classes in one of the randomly selected schools

3. Child signed the child assent form and whose parents/guardian provide written informed consent for the prevalence survey and for the nested trial if hookworm infection was detected.

Trial exclusion criteria were:

1. Child who received benzimidazole drugs such as albendazole and mebendazole within three months before the prevalence survey, to avoid the residual effect of previously administrated drugs.

2. Child who had a history of allergic reaction to any benzimidazole drugs such as mebendazole.

### Trial design, interventions and randomisation

The study aimed to investigate the efficacy of albendazole against hookworm infection with two interventions simultaneously: 1) dual-dose versus single-dose albendazole, 2) taking albendazole with or without fatty food (avocado). A 2x2 factorial outcome assessors-blinded randomised controlled trial was used, as this design has the advantage of enabling investigation of the effects of two independent interventions concurrently and the effect of receiving both interventions together [29]. Participants were therefore randomized in a 1:1:1:1 ratio into one of four arms:

Intervention group 1: participants took 400mg of single-dose albendazole with 200 grams of avocado.

Intervention group 2: participants took dual-dose albendazole (400mg of single dose albendazole for two consecutive days) without taking avocado.

Intervention group 3: participants took dual-dose albendazole (400mg of single dose albendazole for two consecutive days) with 200 grams of avocado on both occasions.

Control group: participants took 400mg of single-dose albendazole without taking avocado.

When positive hookworm cases were identified through the prevalence survey, field workers notified the students of "positive" results. These students were asked to visit the most accessible health centre IV or hospital-level health facility in Mayuge district among four health facilities: 1) Mayuge health centre IV located in Mayuge town council, 2) Kigandalo health centre IV located in Kigandalo sub-county, 3) Kityerera health centre IV located in Kityerera sub-county, and 4) Buluuba hospital in Baitambogwe sub-county. All participants were instructed to have their usual meal on the morning of the visit, following the protocol for later taking albendazole with a fatty food (i.e. in the trial) as outlined in the study of Nagy *et al.* [14]. Each participant was randomised into one group among four treatment options by applying the block randomisation method, stratified by health facility. The randomisation block size was 8, and within each block there were two participants allocated to each of the four trial arms. A randomisation list was prepared beforehand using a random number generator. Each health facility had its own random number list. One or two designated health worker(s) were responsible for the trial in each health centre.

### Drug administration and adverse events

The participants were asked not to take a meal for at least four hours after the drug consumption to exclude the possibility of affecting albendazole absorption by immediately taking foods. During this time, the health personnel in the health centres observed the participants to determine if any adverse events occurred. Water, and other non-caffeinated and non-alcoholic beverages were allowed during the four hours fasting period.

Albendazole (AGOZOLE 400mg; batch number T11098, produced by AGOG Pharmaceutical Ltd., India) was used for the trial. AGOZOLE albendazole is approved by the National Drug Authority (NDA) of Uganda and is widely available in the Ugandan domestic pharmaceutical market (NDA registered number: NDA/MAL/HDP/3013).

For those in the relevant trial arms in our study, we provided 200 grams of avocado, which contains 27–28 grams of fat [30] and is widely available in Uganda at low cost.

For participants in intervention groups 2 and 3, who were taking 400mg of albendazole on two consecutive days, they were asked to return to the same clinic to do so.

A single stool sample was collected from each trial participant three weeks after the treatment to evaluate the albendazole efficacy while avoiding the effect of re-infection after albendazole treatment and egg production by re-infected hookworm.

### Trial outcomes

Trial outcomes were assessed at a follow-up visit 21 days after albendazole treatment. The co-primary outcomes were two commonly-used representative indicators to determine anthelminthic drug efficacy: cure rate (CR) and egg reduction rate (ERR) [31]. CR is defined as the percentage of infected individuals at baseline (i.e. at the time of the prevalence survey) that are free of infection three weeks after treatment. ERR is the difference in the mean eggs per gram before and after the treatment, expressed as a percentage of the mean eggs per gram (EPG) before treatment [32].

### Field and laboratory procedures

Field workers collected stool samples from prevalence survey participants in the randomly selected schools, and from the participants who took part in the albendazole treatment trial three weeks after treatment. The collected stool samples were contained in a designated stool container secured in a cooling box carried by the field workers, and transferred to the district central laboratory in Kigandalo health centre IV where Kato-Katz examination was performed. Two slides were prepared from one stool sample within 24 hours of being delivered to the laboratory. The stool samples not treated with the Kato-Katz method on the same delivery day were contained in the refrigerator with 4˚C condition overnight.

The slides were interpreted by examiners from the Vector-borne and NTDs Control Division (VCD) of the Ministry of Health of Uganda. The supervisor from VCD re-examined 5% of slides to ensure the accuracy of results.

### Sample size

The cure rate of single-dose albendazole for hookworms varies between 36% and 93% in previously reported studies [25–28]. However, Moser *et al* estimated single-dose albendazole drug efficacy was 79.5% in their systematic review [33]. Based on data from Adegnika *et al* [28]; the cure rates for hookworm with single-dose and dual-dose albendazole were 54% (95% CI: 27–81%) and 92% (95% CI: 78–100%). Based on these data, we assumed that the cure rate of single-dose albendazole would be 80% and the cure rate of dual-dose albendazole would be 95%. Regarding the avocado co-administration group, as this was the first clinical trial of its kind, there is no preliminary data to inform sample size calculations for this comparison; we therefore posited an 80% cure rate for the group receiving pure albendazole and a 95% cure rate for the group receiving albendazole in combination with avocado. Therefore, the trial was powered to detect a difference in cure rate of 95% versus 80% for both dual-dose versus single-dose albendazole and for co-administration with fatty foods versus no co-administration.

With type I error probability of 5%, 100 participants in each comparison group were required for 90% power to detect a difference in cure rate of 95% versus 80%. Therefore, the total target sample size for the trial was 186. In order to allow for 5% loss to follow-up, the target sample size was increased to 210.

### Statistical analysis

Data were first recorded in paper-based case report forms (CRF) and then entered on KoBoToolbox for data analysis, with cross-checks by a second investigator to minimize data

transfer errors. Data saved in the KoBoToolbox system were exported into Microsoft Office Excel, and checked for errors, missing values, and extreme values. The cleaned data were exported to Stata SE version 17.0 (Stata Corp; college Station, TX, USA) for statistical analysis. The raw data and the codes to the raw data are included as supplementary information for further studies and systematic review (S1 Data and S1 Code).

Using data from prevalence survey participants, prevalence of STHs with 95% confidence intervals were calculated. The arithmetic mean eggs count was obtained from the microscopic examination of the Kato-Katz method. The intensity of infection was categorised by the criteria of WHO guideline [2]: hookworm, 1–1,999 EPG (light), 2,000–3,999 EPG (moderate), $\geq$4,000 (heavy).

Baseline characteristics including sex, age, anthropometric indicators, co-infections with schistosomiasis, and infection intensity were displayed stratified by each of the four randomization groups. The primary analysis of trial outcomes was done as an "at the margin analysis" [29]. To assess the effect of dual- versus single-dose albendazole, cure rate and ERR were calculated separately for all participants randomized to receive dual-dose albendazole (regardless of whether this was with or without avocado) and compared to cure rate and ERR among all participants randomized to receive single-dose albendazole (regardless of whether this was with or without avocado). Similarly, to assess the effect of co-administration with avocado, cure rate and ERR were calculated for all participants who received albendazole (regardless of dose) with avocado, and compared to cure rate and ERR among participants who received albendazole (regardless of dose) without avocado co-administration. Before conducting the "at the margin" analysis, we assessed whether there was any evidence of effect modification between the two independent interventions, i.e. whether the effect of dual-vs single-dose on outcomes differed in the presence versus absence of avocado, and vice versa. In secondary analysis, the cure rate and ERR for each of the four treatments arms (combinations of dual- versus single-dosing and avocado co-administration versus no co-administration) are also presented.

CR was calculated as the percentage of infected individuals at prevalence survey that are free of infection three weeks after treatment. ERR was calculated as the difference in the mean eggs per gram before and after the treatment, expressed as a percentage of the mean eggs per gram (EPG) before treatment [32]. Both arithmetic and geometric means were used for calculating ERR.

Difference in cure rates between trial arms were assessed using chi-squared tests, and quantified as odds ratios and 95% confidence intervals using Mantel-Haenszel method. Bootstrap re-sampling methods with 1,000 iterations were used to calculate confidence intervals for ERR and the treatment group differences, with differences considered significant when 95% confidence intervals of treatment efficacy excluded the null value. *p*-value lower than 0.05 was considered as statistical significance for all tests.

## Results

### General characteristics of the trial participants

The prevalence survey conducted before the clinical trial detected 352 hookworm-positive cases. Among them, 225 randomly selected students were enrolled in the trial (male:115, female:110). Enrolled students were allocated into four treatments group by applying the block randomisation method. Three students were lost to follow-up between randomisation and outcome assessment at three weeks, the data from the remaining 222 participants were analysed (Fig 1).

The baseline characteristics of participants in each randomisation arm are shown in Table 1. Overall, 51% of participants were male, the mean age was 12.2 years, the mean height

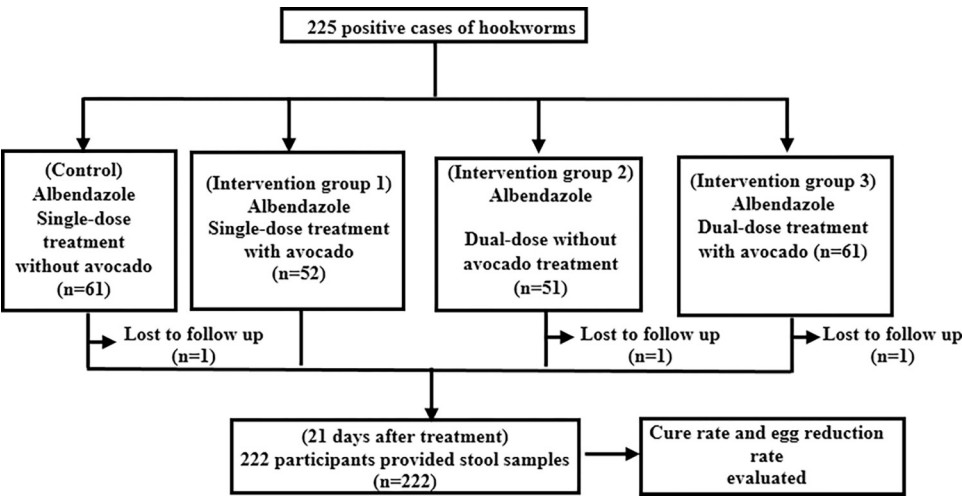

**Fig 1. Study enrolment, randomisation and evaluation chart.**

was 146.6 cm, the mean weight was 36.1 kg, the mean Height for Age z-score (HAZ) was -0.59, and the mean BMI for Age z-score (BAZ) was -0.82. The light intensity infection accounts for 96.9%. Baseline characteristics were generally similar between trial arms, although there was some suggestion of variability (consistent with chance) in the percentage of male participants and the percentage of participants with *S. mansoni* coinfection.

When comparing the characteristics of single-dose albendazole and dual-dose albendazole groups respectively (S1 Table), males account for 52.7% and 49.0%; schistosomiasis co-infection occurred in 17 (15.2%) and 27 (24.6%); hookworm infection was light in 96.4% and 97.3%. Other characteristics of the two groups were similar, including age, height, weight, HAZ and BAZ.

When comparing the characteristics of albendazole without avocado treatment group and albendazole with avocado treatment groups respectively (S2 Table), males account for 50.4%

**Table 1. Baseline characteristics of the clinical trial participants.**

|  | Single-dose albendazole without avocado (n = 60) | Single-dose albendazole with avocado (n = 52) | Dual-dose albendazole without avocado (n = 50) | Dual-dose albendazole with avocado (n = 60) |
|---|---|---|---|---|
| Male n (%) | 28 (46.7) | 31 (59.6) | 29 (58.0) | 25 (41.7) |
| Mean age (SD), years | 12.2 (1.88) | 12.2 (1.61) | 11.9 (1.47) | 12.5 (1.56) |
| Mean height (SD), cm | 145.9 (12.3) | 147.2 (13.9) | 147.6 (12.9) | 145.9 (11.7) |
| Mean weight (SD), kg | 35.1 (5.8) | 36.2 (7.9) | 35.9 (7.5) | 37.3 (7.1) |
| HAZ[a] (SD) | -0.57 (2.67) | -0.49 (2.48) | -0.29 (2.28) | -0.91 (2.19) |
| BAZ[b] (SD) | -0.85 (2.19) | -0.88 (1.45) | -0.94 (1.51) | -0.63 (1.44) |
| *S. mansoni* co-infection, n (%) | 10 (16.7) | 7 (13.5) | 12 (24.0) | 15 (25.0) |
| Hookworm infection intensity, n (%) |  |  |  |  |
| Light (1–1,999 EPG) | 58 (96.7) | 50 (96.1) | 49 (98) | 58 (96.7) |
| Moderate (2,000–3,999 EPG) | 1 (1.7) | 1 (1.9) | 1 (2) | 1 (1.7) |
| Heavy (≥ 4,000 EPG) | 1 (1.7) | 1 (1.9) | 0 (0) | 1 (1.7) |

a height-for-age z-score

b BMI-for-age z-score

**Table 2. Efficacy of dual- versus single-dose albendazole treatment.**

| | Single-dose albendazole (n = 112) | Dual-dose albendazole (n = 110) | Difference between trial arms [OR (95% CI, *p*-value) for cure rates, ERR difference (95% CI, p-value) for ERR] |
|---|---|---|---|
| Number cured | 94 | 106 | |
| Cure rate (95% CI) | 83.9 (75.7–90.2) | 96.4 (90.9–99) | 5.07(1.61–15.96, *p* = 0.002) |
| (arithmetic mean) EPG before treatment (95% CI) | 352.9 (96.79–609.07) | 257.3 (135.76–378.93) | |
| EPG after treatment (95% CI) | 19.5 (-3.72–42.72) | 6.2 (-3.63–16.07) | |
| ERR (95% CI) | 94.5 (76.5–98.9) | 97.6 (91.9–99.8) | 3.1% (95% CI: -3.89–16.39%, *p* = 0.553) |
| (geometric mean) EPG before treatment (95% CI) | 70.08 (53.13–92.45) | 64.96 (49.1–85.95) | |
| EPG after treatment (95% CI) | 1.83 (1.39–2.41) | 1.17 (0.99–1.38) | |
| ERR (95% CI) | 97.4 (96.2–98.2) | 98.2 (97.5–98.7) | 0.8% (95%CI: -0.36–1.98%, *p* = 0.191) |

and 49.6%; schistosomiasis co-infection occurred in 22 (20.0%) and 22 (19.6%); hookworm infection was light in 97.3% and 96.4%. Other characteristics of the two groups were similar, including age, height, weight, HAZ and BAZ.

## Efficacy results

There was no evidence of effect modification between the two trial interventions (dual- versus single-dosing) and co-administration with avocado versus no-administration with avocado (*p* = 0.29). For completeness, descriptive statistics showing the cure rate and ERR in each randomization arm are shown in S3 Table.

The cure rate in the dual dose group was 96.4% (95% CI:90.9–99%), higher than 83.9% (95% CI: 75.7–90.2%) in the single-dose group (OR: 5.07 (95% CI:1.61–15.96), *p* = 0.002) (Table 2).

The arithmetic mean based ERR was 94.5% and 97.6% in the single-dose and dual-dose groups, respectively. The difference of arithmetic mean based ERR between dual-dose and single-dose albendazole was 3.1% (95% CI: -3.89–16.39%, *p* = 0.553). The geometric mean based ERR was 97.4% and 98.2% in the single-dose and dual-dose group. The difference of geometric mean based ERR between single-dose and dual-dose albendazole was 0.8% (95% CI: -0.36–1.98%, *p* = 0.191).

The cure rates among participants taking albendazole without avocado and with avocado were 89.1% and 90.1%, respectively. There was no statistical difference between the two groups (OR: 1.24 (95% CI:0.51–3.03), *p* = 0.622) (Table 3).

**Table 3. Cure rate and ERR between without avocado and with avocado treatment.**

| | Albendazole without avocado (n = 110) | Albendazole with avocado (n = 112) | Difference between trial arms [OR (95% CI, *p*-value) for cure rates, ERR difference (95% CI, p-value) for ERR] |
|---|---|---|---|
| Number cured | 98 | 102 | |
| Cure rate (95% CI) | 89.1 (81.7–94.2) | 90.1 (83.1–94.9) | 1.25 (0.51–3.03, *p* = 0.622) |
| (arithmetic mean) EPG before treatment (95% CI) | 270 (126–413.7) | 340.5 (95.49–585.5) | |
| EPG after treatment (95% CI) | 15.71 (-7.33–38.75) | 10.18 (-0.94–21.29) | |
| ERR (95% CI) | 94.2 (80.1–97.2) | 97 (88.8–99.4) | 2.8% (95% CI: -8.63–14.3%, *p* = 0.629) |
| (geometric mean) EPG before treatment (95% CI) | 69.68 (52.57–92.35) | 65.41 (49.67–86.15) | |
| EPG after treatment (95% CI) | 1.51 (1.19–1.92) | 1.42 (1.14–1.78) | |
| ERR (95% CI) | 97.8 (96.8–98.5) | 97.8 (96.8–98.4) | 0% (95% CI: -1.09–1.13%, *p* = 0.99) |

The arithmetic mean based ERR was 94.2% and 97.0% in the group receiving albendazole without avocado and albendazole with avocado respectively. The difference in arithmetic mean based ERR between the two groups (albendazole without avocado and albendazole with avocado group) was 2.8% (95%CI -8.63–14.3%, p = 0.629). The geometric mean based ERR was 97.8% in both the albendazole without and with avocado groups, with the difference in geometric mean based ERR between the two groups 0% (95%CI: -1.09–1.13%, *p* = 0.99).

## Adverse events

All participants were monitored for four hours after treatment. two participants complained of nausea after taking albendazole, but this symptom subsided without further treatment during the four hours of the observation period. There were no other reported drug-related adverse events from participants in any of the four different treatment arms.

## Discussion

To our knowledge, this is the first clinical trial in Uganda to compare the efficacy of dual-dose versus single-dose albendazole, and the first randomised trial globally to assess the clinical efficacy of albendazole when given with a fatty food against hookworm infection. Our study shows that dual-dose albendazole is more effective in treating hookworm than single-dose albendazole treatment. However, there is no difference in albendazole efficacy when it was co-administered with avocado (a fatty food) compared to without co-administration.

Our findings are consistent with other data which suggest that the cure rate for hookworm is improved with multiple dosing. For instance, the cure rate for hookworm with single-dose albendazole was 54% (95% CI:27–81) in a study in Gabon, versus 92% (95% CI:78–100) with dual-dose [28]. Similarly, a study in China demonstrated a cure rate with single-dose albendazole of 69.1% (95% CI: 55.2–80.9%) versus 92% (95% CI: 80.8–97.8%) with triple-dose albendazole [27]. Multiple-dose albendazole is also recommended for improving its effectiveness against other STHs, especially for *Trichuris trichiura*.

Previous laboratory studies have shown that an active albendazole metabolite (albendazole sulphoxide) plasma concentration is four to six times higher when a fatty food is given with albendazole than when albendazole is taken with water [14,34]. Nagy *et al*.'s study demonstrated a six-fold increase in albendazole sulphoxide plasma concentration when given with a Big Mac, French fries, and a milkshake (57 gram fat, 1,399 kcal) compared to in the group taking albendazole only [14]. This suggests that co-administration of fatty food could increase systemic availability of albendazole sulfoxide [35]. However, there is no clinical study showing the difference in drug efficacy of albendazole against hookworm infection when co-administrating a fatty food, as far as we are aware. Therefore, this clinical study is the first study to compare the drug efficacy of albendazole without a fatty food and of albendazole with a fatty food for treating hookworm infection. While the drug efficacy of albendazole with a fatty food was not better than that of albendazole without a fatty food in our study, further evaluation would be worthwhile. For example, the drug effectiveness of benzimidazoles against other STHs, such as whipworm is known to be low, and fatty food co-administration as an alternative should be explored [36]. *Trichuris trichiura* is not common in Mayuge district and we could not study the effectiveness of albendazole with fatty food co-administration against *Trichuris trichiura* infection.

Drug effectiveness studies comparing different brands of albendazole for treating hookworm in Nepal [15] and Ethiopia [16] demonstrated variable cure rates against hookworm infection, while Belew *et al*. showed that the dissolution rate of albendazole differed by brands [16]. We used the same brand and batch of albendazole for treating all trial participants,

therefore our results will not be influenced by this phenomenon; however, it is possible that the trial results might have differed if an alternative brand had been used. International pharmaceutical companies such as GSK have donated albendazole, commonly used for MDA in many countries. However, the authors used generic albendazole available in Uganda's domestic market since it is not permitted to use the donated GSK albendazole for clinical trial purposes. Albendazole is directly active against helminths in the intestine [37] and a high albendazole dissolution rate increases drug efficacy against hookworm [16]. Therefore, when the active metabolite of albendazole remains in the intestine for a longer duration, it can exert a greater effect on killing intestinal parasites. However, further studies are needed to investigate how the active metabolite can remain in the intraluminal space for an extended period of time.

Preventive chemotherapy is still an essential component for reducing the burden of STHs even though multisectoral and comprehensive interventions, including water, sanitation and hygiene (WASH) improvement, are required to eliminate STHs from a long-term perspective [10]. Albendazole has been used for treating human STHs for a long time. Benzimidazole resistance in animals is common due to the overuse of benzimidazole. In that sense, long-term use of albendazole can potentially develop drug resistance in human STHs. Therefore, considering the drug's effectiveness, the best option for albendazole use is recommended while adopting the WASH program for reducing STHs. At least for hookworm infection, according to our study's result, dual-dose albendazole use is superior to single-dose albendazole considering the cure rate.

There are some limitations of this study. First, the trial was not powered to show the combined effect of dual-dose albendazole with avocado use, which may have high drug efficacy since we use a factorial design in our study. A larger sample size would be required to secure sufficient power to precisely estimate drug efficacy separately among the four treatment groups.

Second, the authors arbitrarily estimated the cure rate of albendazole with the avocado treatment group as 95% since there was no previous similar clinical trial. Therefore, we may not have had sufficient statistical power to detect smaller differences in albendazole efficacy between the without and the with-avocado group.

Third, as described earlier, the amount of fat in 200 grams of avocado might not be enough to increase albendazole efficacy. For example, Nagy *et al.* used 57 grams of fat in their trial and showed six times high in the maximum concentration (Cmax) of albendazole sulphoxide [14]. The 27–28 grams of fat contained in 200 grams of avocado in our study, may be too low in fat. Further work is needed to determine whether higher amounts of fat could increase efficacy of albendazole.

Fourth, the authors cannot explore dual-dose albendazole efficacy and avocado use against moderate- and heavy intensity hookworm infection since 97% of the participants had light intensity infection. Bezie *et al.* displayed the CR and ERR of single-dose albendazole against hookworm infection stratified by infection intensity [38]. The heavy-intensity infection group had 43% CR, while the light-intensity infection group had 94.6% CR. Further studies need to explore the drug efficacy of dual-dose albendazole and fatty meal co-administration against moderate- and high-intensity hookworm infection.

## Conclusion

Single-dose albendazole is the backbone treatment of MDA for hookworm. However, despite the long-term use of single-dose albendazole as an MDA method, when the prevalence of hookworm infection is consistently high, as in Mayuge district in Uganda, the use of dual-dose

albendazole may be a practical recommendation for controlling hookworm infection. This clinical trial indicates that dual-dose albendazole can improve the drug efficacy of albendazole for treating hookworm infection in Mayuge, Uganda. Therefore, we would recommend a change to dual-dose treatment in order to enhance albendazole efficacy in areas where hookworm infection is still prevalent, even though it may be challenging to administer more than a single-day regimen in many settings. In light of the challenges of administering a multi-day regimen, drug combinations such as benzimidazole drugs with ivermectin should also be considered as an alternative means of improving effectiveness while reducing risk of resistance. While we did not demonstrate any benefit with use of avocado co-administration, there remains scope for future studies to investigate whether albendazole efficacy against intraluminal helminth infection could be improved through co-administration with different types of food.

## Supporting information

**S1 Checklist. CONSORT checklist.**
(DOC)

**S1 Protocol. Study proposal for MANE project.**
(DOCX)

**S1 Table. Baseline characteristics by dual-versus single-dose albendazole groups.**
(DOCX)

**S2 Table. Baseline characteristics by avocado versus no avocado groups.**
(DOCX)

**S3 Table. Estimates of cure rate and ERR for the four trial arms.**
(DOCX)

**S1 Data. Raw data of the clinical trial.**
(XLSX)

**S1 Code. Codes to the raw data of the clinical trial.**
(DOCX)

## Acknowledgments

We thank the children who participated in the clinical trial. We thank the medical staff of the Mayuge district who supported this trial. We also thank the staff of the Vector-borne and NTDs control division of the Ugandan Ministry of Health who supervised and conducted stool examinations in the field.

## Author Contributions

**Conceptualization:** Eun Seok Kim, Kalpana Sabapathy, Emily L. Webb.

**Data curation:** Eun Seok Kim, David Lokure.

**Formal analysis:** Eun Seok Kim, Kalpana Sabapathy, Emily L. Webb.

**Funding acquisition:** Eun Seok Kim.

**Investigation:** Eun Seok Kim, Wamboko Aidah, Kabarangira Christine Oseku, David Lokure.

**Methodology:** Eun Seok Kim, Kabarangira Christine Oseku, David Lokure, Kalpana Sabapathy, Emily L. Webb.

**Project administration:** Eun Seok Kim, Kabarangira Christine Oseku, David Lokure.

**Supervision:** Moses Adriko, Kalpana Sabapathy, Emily L. Webb.

**Writing – original draft:** Eun Seok Kim.

**Writing – review & editing:** Eun Seok Kim, Kalpana Sabapathy, Emily L. Webb.

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
