## [Decision Letter · Decision Letter 0]

27 Jan 2023

Dear Kim,

Thank you very much for submitting your manuscript "The impact of dual- versus single-dosing and fatty food co-administration on albendazole effectiveness against hookworm among children in Mayuge district, Uganda: results from a 2x2 factorial randomised controlled trial" for consideration at PLOS Neglected Tropical Diseases. As with all papers reviewed by the journal, your manuscript was reviewed by members of the editorial board and by several independent reviewers. In light of the reviews (below this email), we would like to invite the resubmission of a significantly-revised version that takes into account the reviewers' comments. 

We cannot make any decision about publication until we have seen the revised manuscript and your response to the reviewers' comments. Your revised manuscript is also likely to be sent to reviewers for further evaluation.

Sincerely,

David Joseph Diemert, M.D.

Academic Editor

Cinzia Cantacessi

Section Editor

Reviewer's Responses to Questions

**Key Review Criteria Required for Acceptance?**

**Methods**

-Are the objectives of the study clearly articulated with a clear testable hypothesis stated?

-Is the study design appropriate to address the stated objectives?

-Is the population clearly described and appropriate for the hypothesis being tested?

-Is the sample size sufficient to ensure adequate power to address the hypothesis being tested?

-Were correct statistical analysis used to support conclusions?

-Are there concerns about ethical or regulatory requirements being met?

Reviewer #1: Kim and colleagues study the efficacy of a single versus a double dose of albendazole and whether concomitant fatty food improves the activity of albendazole against hookworm infection. The rationale is not very novel but the data confirm earlier findings. The manuscript is fairly well written. My main concern is that the investigators do not follow many points of the required standards for clinical trials (i.e.GCP)

Reviewer #2: Introduction: please add Strongyloides stercoralis to the "dominant STH", as it was finally included in the list. 

-Lines 59-65: I would add condieratons on WASH and the issue of re-infections if environmental conditions do not change; this also hampers the reduction of STH prevalence.

-Line 78-79: [active metobolite of albendazole; albendazole...]. Did the authors mean to put a simople comma between albendazole and albendazole, to explain that the active metabolite is albendazole sulphoxide? It it not immediately clear now.

- I would state in the Methods that the CONSORT checklist was followed.

-Lines 124-126 (trial exclusion criteria): "to avoid the residual effect of previously adminstered drugs of albendazole": I would delete "of albendazole". Criterion 2: I would change as "Child who had a hisotry of allergic reaction to any benzimidazole drugs"

-TRial design, interventions and randomization paragraph, lines 145-146: it is not clear to me why the participants were instructed to have a fatty meal while the investigators wanted to standardize the fat introduction depending on the randomization arm. Were all participatins invited to have a fatty meal before reaching the study site? Why?

-Lines 150-151: where was the randomization list kept? Who kept it?

Drug administration and adverse events paragraph. I think that ilnes 162-164 better fit the Discussion rather than the methods

**Results**

-Does the analysis presented match the analysis plan?

-Are the results clearly and completely presented?

-Are the figures (Tables, Images) of sufficient quality for clarity?

Reviewer #1: Abstract: delete this sentence as it does not fit in this context (the activity observed was not worrisome): There is growing concern about the low effectiveness of single-dose albendazole, for which there are several possible explanations, including drug resistance and lower absorption using drugs from certain manufacturers.

Introduction: 

Line 48: I would provide the most recent DALY figures or at least a range

Line 56: the albendazole dose for 1-2 year old children is 200 mg

Line 59: where are the prevalences still high? I would shift this entire paragraph (line 59-65) after the following paragraph (lines 66-75)

Line 92-96: was the trial conducted according to GCP? It seems important elements of the GCP practice, as monitoring etc. are missing.

Line 113-115: I would delete the design here as it is mentioned below

Line 145: if the participants had already a meal in the morning prior to the study medication how could you control for a food effect 

Line 162-166: this does not fit here, as already mentioned in the discussion it can be deleted here

Line 169-173: explanation not needed as 14-21 days is the recommended period to take follow up samples. 

Line 182: how many samples were collected from each participant?

Line 206: ODK/Excel should not be used in clinical trials as no audit trail

Results

 Line 256: were

Line 263-274: all data is shown in the table and hence this paragraph can be considerably shortened 

Line 276 line 288 and line 305: efficacy

Discussion

Line 322: not necessarily, as for intestinal nematodes gut concentrations are more important than plasma levels, so whether a co-administration of fat or an activity against T. trichiura will be observed is questionable

Line 331 and following: can be deleted here, as already mentioned in the limitations below

Line 341: I would add here that you used a generic product and not the most commonly used GSK product

Line 349: preventive chemotherapy is based on a single dose for practicality reason, how would you administer a double dose?

Line 367/368: delete, this will be part of a data availability statement

The references should be revised (often page numbers etc. missing)

The study protocol is not sufficient for a clinical trial, e.g. study synopsis, investigators, insurance, monitoring etc. are missing

Consort checklist missing

Reviewer #2: Table 1 - I would report the acronyms used in the table as footnotes. HAZ is missing also from line 266 

Effectiveness of results: actually in a clinical trial I would talk about efficacy rather than effectiveness. Main results should be reported either in the Tables or in the text, otherwise are redoundant. Tables 2 and 3: I think that adding a column reporting OR and p values would be better than reporting them in a line, unrelated to the values they are comparing.

**Conclusions**

-Are the conclusions supported by the data presented?

-Are the limitations of analysis clearly described?

-Do the authors discuss how these data can be helpful to advance our understanding of the topic under study?

-Is public health relevance addressed?

Reviewer #1: (No Response)

Reviewer #2: The conclusions are supported by the results. I would add some further considerations:

- For increasing effectiveness and reducing emergence of resistance, combination of benzimidazole drugs with ivermectin is also considered. This combination is already present in the WHO model list

-For MDA, it is difficult to administer more than a single day regimen in many settings. Also in light of this, drug combination would have advantages compared to increaseing doses of albendazole/mebendazole

Line 369: from here on, the Authors draw the main conclusions. I would hence add the "conclusions" title.

**Editorial and Data Presentation Modifications?**

Reviewer #1: (No Response)

Reviewer #2: (No Response)

**Summary and General Comments**

Reviewer #1: (No Response)

Reviewer #2: The relevance of this study is mainly on the fatty meal component, as the reduced efficacy of a single administration of albendazole, in specific settings, is quite known. The trial was well conducted and the results are reported clearly, although some revisions should be done in order to make the text less redoundant (considering the Tables). Alghough the results here point out at an increased efficacy of multiple doses of albendazole, in the Discussion I would point out that giving multiple doses in a MDA context is not always feasible, hence other strategies should hance be explored (such as combination with fatty meal and drug combinations)

PLOS authors have the option to publish the peer review history of their article (what does this mean?). If published, this will include your full peer review and any attached files.

Reviewer #1: No

Reviewer #2: No
---

## [Decision Letter · Decision Letter 1]

8 May 2023

Dear Kim,

Thank you very much for submitting your manuscript "The impact of dual- versus single-dosing and fatty food co-administration on albendazole effectiveness against hookworm among children in Mayuge district, Uganda: results from a 2x2 factorial randomised controlled trial" for consideration at PLOS Neglected Tropical Diseases. As with all papers reviewed by the journal, your manuscript was reviewed by members of the editorial board and by several independent reviewers. The reviewers appreciated the attention to an important topic. Based on the reviews, we are likely to accept this manuscript for publication, providing that you modify the manuscript according to the review recommendations. 

Sincerely,

David Joseph Diemert, M.D.

Academic Editor

Cinzia Cantacessi

Section Editor

Reviewer's Responses to Questions

**Key Review Criteria Required for Acceptance?**

**Methods**

-Are the objectives of the study clearly articulated with a clear testable hypothesis stated?

-Is the study design appropriate to address the stated objectives?

-Is the population clearly described and appropriate for the hypothesis being tested?

-Is the sample size sufficient to ensure adequate power to address the hypothesis being tested?

-Were correct statistical analysis used to support conclusions?

-Are there concerns about ethical or regulatory requirements being met?

Reviewer #1: The authors have done a good job revising the manuscript. I would still suggest using the term efficacy, as often trial data is reported and not as the authors point out, real world scenario (e.g. line 94 introduction). The authors added as much as possibility on the GCP compliance though there are several small issues (e.g. the use of Excel) which should be fixed in future trials. I would also rewrite the argumentation the absorption/fat hypothesis as summarized below 

Abstract 

Could be merged with previous sentence: The difference of ERR between the two groups (with and without avocado) was 2.8% (95% CI -8.63-14.3%, p=0.629).

Author summary

Last sentence, I am not convinced as it might not require an improved absorption but rather dissolution

Introduction

Line 89: albendazole

 Line 112: council

M&M

Line 212: the cure rate assumption 50% versus 70% is very optimistic—a single dose of albendazole has quite a good efficacy and I am surprised by the 54% value. At the end the difference was around 10%-was the study underpowered? This is not mentione din the limitation section. What were the assumptions for the avocado arms?

Discussion

Line 355: why would it not be permitted to use the GSK drug? In worst case it could have been purchased. I would delete this here. 

Line 357-360 it is not really clear how this argument relates to the different brands of tablets. Would this not rather belong to the discussion on the fatty food?

Line 379: is it really worth to conduct further studies against Trichuris and with a higher fat dose? At the end it is pure speculation and there is no evidence why the drug should perform better in the presence of fat. We do not know what is driving efficacy, e.g. intrgastrical concentrations or plasma levels. 

Similar on line 399—there is no correlation with regard to systemic availability and efficacy from PK/PD studies, so I would challenge your last sentence that due to the biological rationale for improved absorption with high fat levels.

Reviewer #2: (No Response)

**Results**

-Does the analysis presented match the analysis plan?

-Are the results clearly and completely presented?

-Are the figures (Tables, Images) of sufficient quality for clarity?

Reviewer #1: trial raw data should be provided (egg counts etc)

Reviewer #2: The only point that I raised in my previous review and was not addressed concerned the format of Tables 2 and 3: I suggested to add a column reporting the results of the statistical tests (OR and p) for clarity. I still believe that would be better than having OR and p in a line

**Conclusions**

-Are the conclusions supported by the data presented?

-Are the limitations of analysis clearly described?

-Do the authors discuss how these data can be helpful to advance our understanding of the topic under study?

-Is public health relevance addressed?

Reviewer #1: (No Response)

Reviewer #2: (No Response)

**Editorial and Data Presentation Modifications?**

Reviewer #1: (No Response)

Reviewer #2: (No Response)

**Summary and General Comments**

Reviewer #1: (No Response)

Reviewer #2: I am satisfied by the authors' reply and revisions. As I pointed out previously, the only point that I still have concerns the Tables 2 and 3.

PLOS authors have the option to publish the peer review history of their article (what does this mean?). If published, this will include your full peer review and any attached files.

Reviewer #1: No

Reviewer #2: No

Figure Files:

Data Requirements:

Reproducibility:

References

---

## [Decision Letter · Decision Letter 2]

6 Jun 2023

Dear Kim,

We are pleased to inform you that your manuscript 'The impact of dual- versus single-dosing and fatty food co-administration on albendazole efficacy against hookworm among children in Mayuge district, Uganda: results from a 2x2 factorial randomised controlled trial' has been provisionally accepted for publication in PLOS Neglected Tropical Diseases.

Best regards,

David Joseph Diemert, M.D.

Academic Editor

Cinzia Cantacessi

Section Editor

Reviewer's Responses to Questions

**Key Review Criteria Required for Acceptance?**

**Methods**

-Are the objectives of the study clearly articulated with a clear testable hypothesis stated?

-Is the study design appropriate to address the stated objectives?

-Is the population clearly described and appropriate for the hypothesis being tested?

-Is the sample size sufficient to ensure adequate power to address the hypothesis being tested?

-Were correct statistical analysis used to support conclusions?

-Are there concerns about ethical or regulatory requirements being met?

Reviewer #1: (No Response)

**Results**

-Does the analysis presented match the analysis plan?

-Are the results clearly and completely presented?

-Are the figures (Tables, Images) of sufficient quality for clarity?

Reviewer #1: (No Response)

**Conclusions**

-Are the conclusions supported by the data presented?

-Are the limitations of analysis clearly described?

-Do the authors discuss how these data can be helpful to advance our understanding of the topic under study?

-Is public health relevance addressed?

Reviewer #1: (No Response)

**Editorial and Data Presentation Modifications?**

Reviewer #1: (No Response)

**Summary and General Comments**

Reviewer #1: (No Response)

PLOS authors have the option to publish the peer review history of their article (what does this mean?). If published, this will include your full peer review and any attached files.

Reviewer #1: No

---

## [Editor Report · Acceptance letter]

19 Jun 2023

Dear Kim,

We are delighted to inform you that your manuscript, "The impact of dual- versus single-dosing and fatty food co-administration on albendazole efficacy against hookworm among children in Mayuge district, Uganda: results from a 2x2 factorial randomised controlled trial," has been formally accepted for publication in PLOS Neglected Tropical Diseases.

Best regards,

Shaden Kamhawi

co-Editor-in-Chief

Paul Brindley

co-Editor-in-Chief
